# Constraining Latent Space to Improve Deep Self-Supervised e-Commerce Products Embeddings for Downstream Tasks

## Abstract

The representation of products in a e-commerce marketplace is a key aspect to be exploited when trying to improve the user experience on the site. A well known example of the importance of a good product representation are tasks such as product search or product recommendation. There is however a multitude of lesser known tasks relevant to the business, examples are the detection of counterfeit items, the estimation of package sizes or the categorization of products, among others. It is in this setting that good vector representations of products that can be reused on different tasks are very valuable. Past years have seen a major increase in research in the area of latent representations for products in e-Commerce. Examples of this are models like Prod2Vec or Meta-Prod2Vec which leverage from the information of a user session in order to generate vectors of the products that can be used in product recommendations. This work proposes a novel deep encoder model for learning product embeddings to be applied in several downstream tasks. The model uses pairs of products that appear together in a browsing session of the users and adds a proximity constraint to the final latent space in order to project the embeddings of similar products close to each other. This has a regularization effect which gives better features representations to use across multiple downstream tasks, we explore such effect in our experimentation by assessing its impact on the performance of the tasks. Our experiments show effectiveness in transfer learning scenarios comparable to several industrial baselines.

## 1 Introduction

The e-Commerce environment has been growing at a fast rate in recent years. As such, new tasks propose new challenges to be resolved. Some key tasks like product search and recommendation usually have large amounts of data available and dedicated teams to work on them. On the other hand, some lesser known but still valuable tasks have less quality annotated data available and the main goal is to resolve them with a small investment. Examples of the latter are counterfeit/forbidden product detection, package size estimation, etc. For these scenarios, the use of complex systems is discouraged in favor of industry proven baselines like bag-of-words or fastText (Joulin et al., 2016).

In particular, with the advent of "Feature Stores" (Li et al., 2017), industrial applications are seeing a rise in the adoption of organization-wide representations of business entities (customers, products, etc.). These are needed in order to speed up the process of building machine learning pipelines to enable both batch training and real-time predictions with as low effort as possible.

In the present work we explore the representation learning of marketplace products to apply in downstream tasks. More specifically we aim to train an encoder that that can transform products into embeddings to be used as features of a linear classifier for a specific task, thus avoiding feature engineering for the task. The encoder model training is done is a self-supervised fashion by leveraging browsing session data of users in our marketplace. Using product metadata and an architecture inspired on the recent work of Grill et al. (2020), we explore how the use of pairs of products in a session can enable transfer learning into several downstream tasks. As we discuss further in Section 3, we extend on the work of Grill et al. (2020) with a new objective function that combines their

original idea with a cross entropy objective. Our experiments show that the added objective helps the model converge to better representations for our tasks.

We also show, through experimental evaluation, that the encoder model learns good representations that achieve comparable results with several strong baselines including fastText (Joulin et al., 2016), Meta-Prod2Vec (Vasile et al., 2016), Text Convolutional Networks (Kim, 2014) and BERT (Devlin et al., 2018) in a set of downstream tasks that come from some of our industrial datasets.

This paper is structured as follows: Section 2 presents other works in the area of product representation and also the works we take inspiration to design the encoder model and establish how our approach differs from the previous literature. Section 3 describes in detail all the components of our proposed architecture. Section 4 lists our experimental evaluation setup. Section 5 shows the results of our experimentation. Finally, in Section 6 we summarize our findings and delimit our line of future work.

## 2 BACKGROUND

Recent years have seen a dramatic increase of latent representations, which have proven to be more relevant in transfer learning scenarios in Computer Vision with the aid of large pre-trained models (Raina et al., 2007; Huh et al., 2016); and, more recently, with the aid of architectures for training unsupervised language models like LSTMs (Merity et al., 2017) or the attention mechanism (Vaswani et al., 2017), transfer learning has seen an explosion of applications in Natural Language Processing (Howard & Ruder, 2018; Devlin et al., 2018; Radford et al., 2018). For the case of the e-commerce environment, there is extensive research work in the area of latent representation for some of the main tasks.

In the area of recommender systems there is a very large body of work in which the idea is to use information of the user shopping session to generate latent representations of the products. The Prod2Vec algorithm (Grbovic et al., 2015) proposed the use of word2vec (Mikolov et al., 2013) in a sequence of product receipts coming from emails. The Meta-Prod2Vec algorithm (Vasile et al., 2016) extended upon Prod2Vec by adding information on the metadata of a user shopping session.

Using metadata of the products during a stream of user clicks is explored with the aid of parallel recurrent neural networks (Hidasi et al., 2016) where the authors use images and text of an product to expand in order to have richer features to model the products in the session.

Other works that uses more metadata, in this case the user review of an product, is DeepCoNN (Zheng et al., 2017), which consists of two parallel neural networks coupled in the last layers. One network learns user behaviour and the other learns product properties, based on the reviews written.

There is also work in the area of modelling information on session-aware recommender systems (Twardowski, 2016), where the user information is not present and the focus of the task is leaning towards using the session information to recommend products.

This extensive research of representation learning for marketplace products is heavily influenced with the end goal of recommendation. Many of them also leverage from the unsupervised information available such as sessions, reviews, metadata, etc.

In this work the end goal of the representations is not recommendations, but different downstream tasks that we have available from challenges we face in our marketplace. For that we propose a deep encoder architecture that follows the work presented in "Bootstrap Your Own Latent" (BYOL) (Grill et al., 2020) with the intended objective of learning embeddings of products of the same session close to each other in the latent space. However, our experiments showed that this was not enough to ensure the transfer of knowledge, as such we extended the learning objective of BYOL to have a cross entropy objective using the product category as target and we explore how the correct combination of each part of the objective function impacts on the quality of the final embeddings.

The main contributions of our paper are the following: 1) a novel deep encoder architecture that can be trained on pairs of products found in user browsing sessions, 2) a study of how this architecture performs for downstream tasks compared to some strong proposed baselines, 3) an extension to the BYOL architecture to a different domain from the proposed by Grill et al. (2020) and how it impacts on the final results.

## 3 METHOD

### 3.1 BROWSING SESSION DATA

The data needed to train our Product Embeddings are browsing sessions of different users in the marketplace. Each product a user visits (i.e. check the details) is part of the browsing session of the user. A session ends when at least $\mathbf{T}$ minutes have passed without new visits.

More formally, given a sequence of products $s = (p_1, ..., p_n)$ where $p_i$ is an product and $T(p_i)$ is the timestamp of the accessed product, we have that $T(p_{i+1}) - T(p_i) \leq \mathbf{T}$ for a fixed time window $\mathbf{T}$ in minutes. Each product $p_j$ in the session is represented by two attributes: the title $t(p_j)$ and the category $c(p_j)$. The title of the product is written by the user of the marketplace. It contains a brief description of the product with some information such as brand, model, measures, etc. It depends on the product itself, and the same product can appear with different titles. Some products have an associated id to identify that they are the same product sold by different users, but that is not always the case. As the language of our marketplace is Spanish, the titles are normalized by stripping accents, removing stopwords and punctuation, lowercase of words and whitespace normalization. The category of the product is selected by the user from a list of fixed categories. These make both attributes susceptible to noise as are driven by the user's actions (e.g. are spelling errors on the title or missing categories for one-of-a-kind products).

We use the titles of these products as their appear in the sessions to build a sentence piece tokenizer (SPM) (Kudo & Richardson, 2018). This same tokenizer is applied to the titles to reduce out-of-vocabulary related issues. The titles are finally represented to the model as a sequence of sub-word tokens from the SPM.

We see all the possible combinations without replacement with two products between all the products of a session. Each product pair is what we feed to our model. The main concept here is that we are interested in products that are part of the same session, regardless of the order they were accessed by the user. We also remove all duplicate products in a single session (since it is common for the user to access the same product more than once in the time frame).

### 3.2 PROPOSED MODEL

We are interested in finding an encoder to map the product title to a latent space that minimizes the distance of embeddings of a product in the same session while maximizes the distance between products of different sessions. Once we have the encoder function, it is used on the product titles of downstream tasks where the session information is not available. We transfer knowledge from the sessions into the tasks and train a linear model for the task.

More formally, given an encoding function $f_\theta$ we use the information of the browsing session to calculate the parameters $\theta$. Then given a task dataset $S = \{(x_1, y_1), ..., (x_n, y_n)\}$, we use $f_\theta$ to encode $\{f_\theta(x_1), ..., f_\theta(x_n)\}$ and use those features to train a linear classifier $g_\omega$ on the weights $\omega$ while freezing the weights $\theta$.

#### 3.2.1 BOOTSTRAP YOUR OWN LATENT

To constraint the latent space in order to minimize the distance between products of the same session while maximizing the distance with products of other sessions we originally aim at the idea of unsupervised visual representation learning methods such as SimCLR (Chen et al., 2020) and MoCo (He et al., 2020). Both these methods rely on contrasting positive and negative examples in order to learn better representations, finding good negative examples in our environment was non-trivial. We then came across "Boostrap Your Own Latent" (BYOL) (Grill et al., 2020) which expanded the previous architectures with an interesting concept that didn't require the explicit sampling of negative examples and we based our architecture on it[1].

---

[1]We need to point out that according to `https://untitled-ai.github.io/understanding-self-supervised-contrastive-learning.html` the reason why BYOL works is an implicit contrastive learning given by the use of batch normalization. We haven't done an extensive research but some preliminary experiments did show that removing batch normalization had a negative effect on the overall performance of the representations.

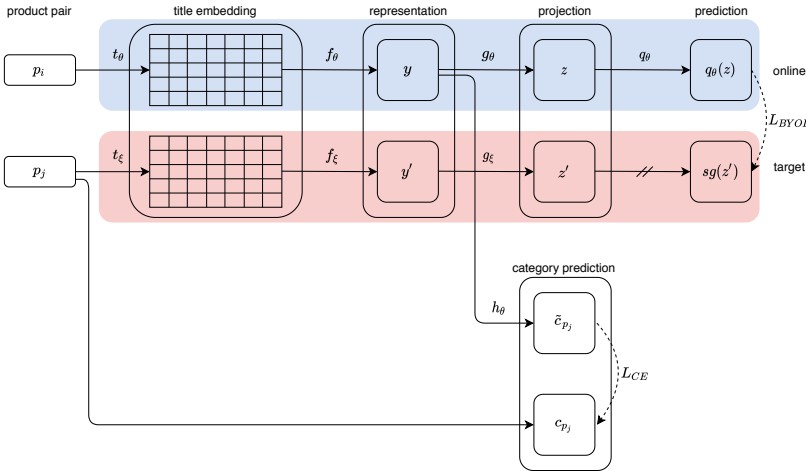

Figure 1: The model's architecture is based on BYOL (Grill et al., 2020). The model receives as input a pair of products $p_i$ and $p_j$. One product is projected through the *online* network until obtained the projector value $q_\theta(z)$. The other product is projected through the *target* network until obtained the value $z'$. The projection $y$ of $p_i$ is also used to obtain the predicted category $\tilde{c}_{p_j}$ of $p_j$ with aid of the category predictor $h_\theta$. The model is trained on the weights $\theta$ by minimizing the objective functions $L_{BYOL}$, which is the distance between $q_\theta$ and $sg(z')$ (where $sg$ means stop-gradient) and the cross entropy loss $L_{CE}$ between the predicted category $\tilde{c}_{p_j}$ and the category $c_{p_j}$ of $p_j$.

### 3.2.2 MODEL'S ARCHITECTURE

Figure 1 shows the model architecture. The core idea of the model is to learn a representation $y$ that can be used for downstream tasks. As it is shown in the Figure, the base of our architecture is heavily inspired by the model proposed in Grill et al. (2020).

In the original architectures of SimCLR, MoCo and BYOL, the input of the model is a single image that is augmented with different views of it. This is the self-supervision given by image itself. In our scenario, our main hypothesis is that different products of the same browsing session can be thought as different augmentations of the same session.

In our original set of experiments, the products were solely represented by their title. The generation of the representation $y$ depends on the encoder we apply to obtain a feature vector of the title of the product. In our setup this encoder is the Text Convolutional Neural Network of Kim (2014) as experiments show that the nature of these networks function very well with the type of text data we have.

More formally, given a pair of products that are part of the same session: a *key* product $p_i$ and a *query* product $p_j$, the BYOL model consists on two paths: the *online* network and the *target* network. The online network is defined by a set of weights $\theta$ and is comprised of five stages: an *embedding lookup* $t_\theta$, a *representation encoder* $f_\theta$ (in our case a Text CNN), a *projector* $g_\theta$, and a *predictor* $q_\theta$.

The target network has a very similar architecture to that of the online network, except for the predictor $q_\theta$, it also has a different set of weights $\xi$. The weights of the target network are not optimized with the data, but are obtained by a slow exponential moving average of the online parameters $\theta$, given a target decay rate $\tau \in [0, 1]$, after each training step the weights $\xi$ are updated: $\xi \leftarrow \tau\xi + (1 - \tau)\theta$. Both $\theta$ and $\xi$ are initialized to the same values when creating the networks.

After training the model, we are interested in the encoder $f_\theta \circ t_\theta$ that gives us the feature vector $y$ to use on downstream tasks. The rest of the model is discarded.

### 3.2.3 BYOL OBJECTIVE

The approach proposed by Grill et al. (2020) is to train the online network by minimizing the distance between the prediction $q_\theta$ and the projection $z'$ of the target network. As seen in Figure 1, with the

use of the stop-gradient function "sg", the idea is to avoid optimizing over the weights $\xi$ of the target network. This is to avoid the collapse given by the trivial solution of output a constant value for the embedded space. The weights $\xi$ of the target network are obtained instead by a slow-moving average of the weights $\theta$ of the online network.

From the key product $p_i$, the online network outputs a *representation* $y = f_\theta(t_\theta(p_i))$ and a *projection* $z = g_\theta(y)$. The target network outputs from the query product $p_j$ the *target representation* $y' = f_\xi(t_\xi(p_j))$ and the *target projection* $z' = g_\xi(y)$. In the final stage, the online network outputs a *prediction* $q_\theta(z)$ of $z'$. The final step is to take the loss between the $L_2$-normalized versions of the predictions $q_\theta(z)$ and the target projections $z'$:

$$L_{BYOL} = 2 - 2 \cdot \frac{\langle q_\theta(z), z' \rangle}{\|q_\theta(z)\|_2 \cdot \|z'\|_2} \tag{1}$$

### 3.2.4 CATEGORY PREDICTION

In our original setup we explored the use of the BYOL optimization objective alone. However we found it difficult in some scenarios to really achieve knowledge transfer to some of the tasks we had. As such, we decided to take advantage of some of the others meta-data available for the products. In particular, for this work we are using the product category which is one of the most common meta data available at our marketplace.

Inspired by the idea of Meta-Prod2Vec (Vasile et al., 2016) we added an extra optimization step to the original BYOL architecture. In this case, we add a new predictor $h_\theta$ that maps the representation of the key product $p_i$ to the category $c_{p_j}$ of the query product $p_j$: $\tilde{c}_{p_j} = (h_\theta \circ f_\theta \circ t_\theta)(p_i)$. Then we optimize the cross entropy loss:

$$L_{CE} = -\log\left(\frac{e^{\tilde{c}_{p_j}}}{\sum_l^C e^{\tilde{c}_l}}\right) \tag{2}$$

### 3.2.5 FINAL OPTIMIZATION OBJECTIVE

Given that the losses in Eq. 1 and Eq. 2 only refer to one way of the product pair $p_i, p_j$, we can make the final loss of the model symmetric by swapping the key and the query products, and give them as input to the networks using $p_j$ as the key and $p_i$ as the query:

$$L = (L_{BYOL_{p_i}} + L_{BYOL_{p_j}}) + (L_{CE_{p_i}} + L_{CE_{p_j}}) \tag{3}$$

### 3.2.6 IMPORTANCE OF EACH OBJECTIVE

The question that arises is how important each optimization objective is to the overall performance of the representations learnt. A first approximation is to sum them as in Eq. 3. However, our experiments show that this is not the best case. Moreover, one hypothesis we originally had is that the transfer of knowledge was solely obtained because we added the cross entropy objective $L_{CE}$. Our experimental results however show a different story. To see the importance of each part of the loss in the final performance, we added an extra hyperparameter $\alpha \in [0, 1]$ that weights the final objective in a complementary fashion:

$$L = \alpha(L_{BYOL_{p_i}} + L_{BYOL_{p_j}}) + (1 - \alpha)(L_{CE_{p_i}} + L_{CE_{p_j}}) \tag{4}$$

Successive experiments showed us that at larger values of $\alpha$, with more focus on the $L_{BYOL}$ objective we had better representations for the downstream tasks. However, if $\alpha$ takes the value of 1, completely cancelling the influence of $L_{CE}$, the representations cannot achieve good results.

## 4 EXPERIMENTAL EVALUATION

### 4.1 DATASETS

The experiments need two kind of datasets: one for the self-supervised training of the Product Embeddings and the others for the downstream tasks. For better validation of our hypothesis, we experimented with data coming from two countries where our marketplace is present. Each country has it own sets of session data and downstream tasks data, although the objective is the same, the data distribution is not. For the scope of this work we present 4 binary classification tasks:

**Counterfeit Products** A dataset based on the users reported counterfeit objects that are being sold in our marketplace.

**Forbidden Products** A dataset of products that are prohibited to commercialize in our marketplace.

**Free Shipment Eligible** A dataset with products sizes (height, length and width). The task is to determine if the product is eligible for free shipment, this happens when all of the dimensions are smaller than 70 cm.

**Product Condition** A binary classification to check if the product is new or used.

The tasks datasets were divided into train, test and validation with a 60/20/20 split. The validation set is for hyperparameter tuning and the final results are taken from the test set. We are interested in how the model works when there is a limited data availability. For this we run experiments on subsamples of the train data, using 100, 500, 1000, 5000, 10000, 50000 and 100000 randomly chosen instances. To assess the impact of selecting the data, we run experiments on 5 samples for each training size. We report the results of mean and standard deviation for each training sample on each task. These tasks are represented by the title and the label of the task. The titles are pre-processed with the same text normalization as for the sessions dataset.

**Note on the data** Due to the sensitive nature of these datasets, we are working on the anonymization and future release of them. We cannot guarantee the availability at reviewing time, but we are doing our best efforts in order to have them available along the code to reproduce the experiments for the camera ready.

### 4.2 MODELS

The self-supervised encoder $f_\theta \circ t_\theta$ is used to extract the representations of each of the tasks products and we use those representations with a logistic regression classifier. We use the validation data to select the $\alpha$ value (we assess the impact using 5 values of $\alpha$: 0, 0.1, 0.5, 0.9 and 1). Due to space limitations we leave the detailed hyperparameters for the reader to see in Appendix A.1.

We compare the with the following baselines: fastText, Text CNN, Spanish BERT with logistic regression, and Meta-Prod2Vec with K Nearest Neighbors. For a more detailed explanation of how the baselines are applied please refer to Appendix A.2.

The metric used for comparison of performance is the Area Under ROC (or AUCROC), which measures the performance of the classifier being independent of the threshold.

## 5 RESULTS AND DISCUSSIONS

### 5.1 IMPACT OF THE LOSSES IN THE FINAL REPRESENTATION

Figure 2 shows the learning curves that measure the impact of the selection of the $\alpha$ hyperparameter that controls the importance of each of the losses in Eq. 4. Each graphs represents a task for each site. The $x$ axis represents the size of the training set and the $y$ axis represents the the performance on the validation set measured by AUCROC. The graphs on the left represent Site A and the ones on the right represent of Site B. The color intensity and the style of the line represents the value of $\alpha$. A higher value of $\alpha$ is represented by a darker shade of blue. As we can observer, in almost all plots the pattern is clear, the better performance is for the value of $\alpha$ of 0.9 specially at very low training samples when the curve is more steep for cases of $\alpha$ equal to 0.9. However when we reach the value

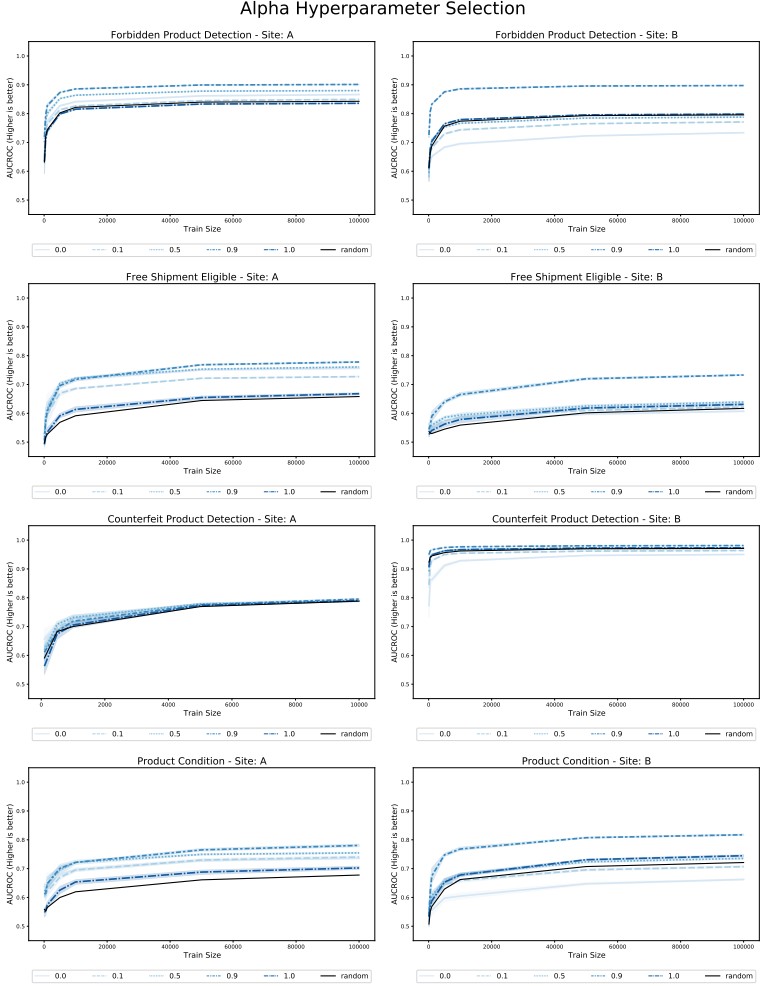

Figure 2: Impact of weighting each loss differently in the performance of the models for the downstream tasks. The darker the color, the higher the $\alpha$. For the black line, the encoder is random (i.e. no self supervised training). The $x$ axis represents the amount of training data. The $y$ represents the AUCROC at different training sizes.

of $\alpha$ equal to one the model collapses. For comparison sake we also plot the value of a random encoder (i.e. not trained with the self-supervised task), represented by the black line. It is to notice that in some cases, a lower value of $\alpha$ or a value of $\alpha$ equal to 1, the model performs worse than random. This results support the hypothesis that both $L_{BYOL}$ and $L_{CE}$ are essential for the model to transfer knowledge, and showcase the importance of $L_{BYOL}$ which is the part of the objective function in charge of shortening the distance between products of the same session.

## 5.2 COMPARISON WITH BASELINES

Figure 3 presents the comparison of our model (named "Product Embeddings" in the plot) to the baselines defined in previous sections. Similarly to the case of Fig. 2, the $x$ axis represents the size of the training set and the $y$ axis represents the performance measured by AUCROC. In this case, however, the performance is measured over the test set of each task. Each line of different color represents the model. The shade around the lines represents the confidence interval since each model was run on 5 different subsamples of each training size. The plots show a clear advantage of more complex models like TextCNN and Spanish BERT, specially as the amount of data available increases. However our model does not fall behind, specially since the tuning of hyperparameters

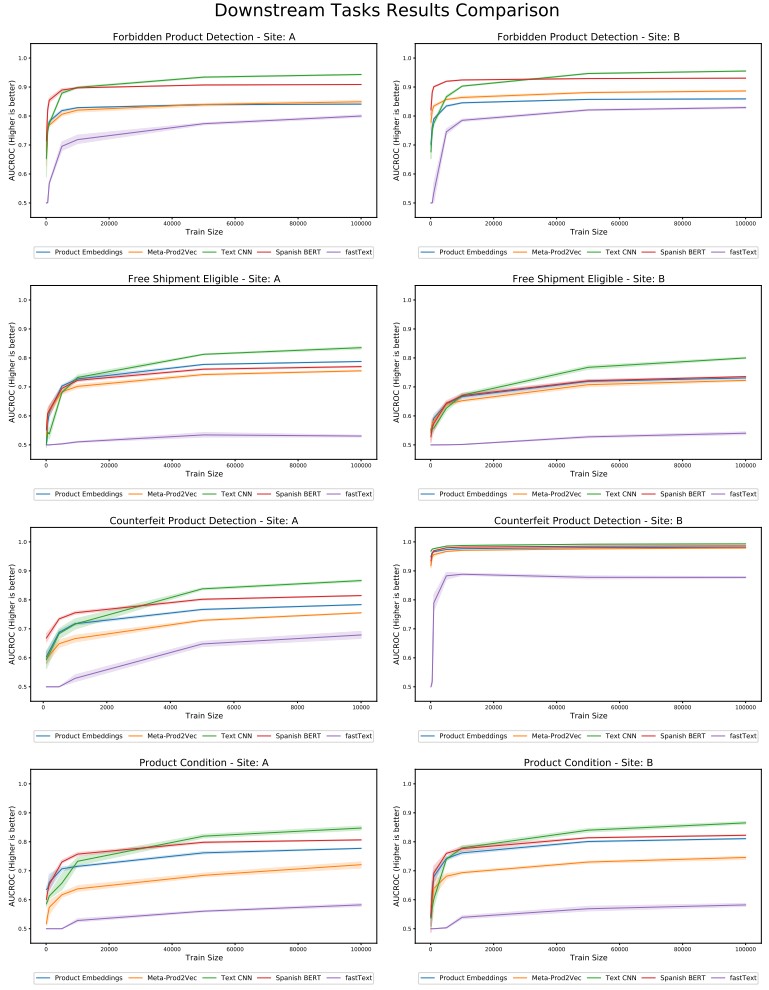

Figure 3: Results of the experiments. Each graph represents the performance comparison in different downstream tasks. The different models are presented the different colors. The $x$ axis represents the amount of training data. The $y$ axis, depending on the task, represents results using different metrics. The $y$ axis depends on the task, represents the metric associated with the task.

was minimal. This is important since TextCNN requires GPU for training on each task, and BERT require GPU at inference time. Our model was trained on GPU for the self-supervised task, but the downstream task the model is only used for inference and does not require a GPU to do so. On the other hand the model outperforms both Meta-Prod2Vec and specially fastText. The case of Meta-Prod2Vec is particularly important since our model achieves better results by only encoding based on the title, without the need of generating one embedding per product nor any other metadata at inference time.

# 6 CONCLUSIONS

In this work we presented a novel deep encoder model trained on self-supervised fashion from a marketplace environment and used as a feature extractor for downstream tasks. We extended the work of Grill et al. (2020) to a new domain and proved that, with some adjustments, is a feasible solution worth exploring. The model shows good performance in comparison to some industrial baselines with advantages over many of them. Future work will include the exploration of different learning objectives with aid of other metadata present in the items.

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

## A EXPERIMENTAL SETUP HYPERPARAMETERS

### A.1 ARCHITECTURE AND MODEL TRAINING

The selected architecture was an embedding lookup $t_\theta$ for 30,000 tokens of the vocabulary of the sentence piece tokenizer and dimension 512.

The representation encoder $f_\theta$ is a Text CNN with 4 kernels of size 2, 3, 4 and 5 tokens (as these are 1D CNN), a total of 128 filters for each kernel size with a global max pooling operation. The output dimension of the encoder is 512.

The projection encoder $g_\theta$ is a *multi-layer perceptron* (MLP) that takes the 512 and expands it to a hidden layer of size 1025, followed by batch normalization Ioffe & Szegedy (2015) with a decay rate of 0.9, rectified linear unit Nair & Hinton (2010) and a final linear layer of size 128. The predictor $q_\theta$ uses the same architecture of the projector. Similarly, the category predictor $h_\theta$ follows the same architecture previously defined, except for the final linear layer which will project to the cardinality of the set of product categories. The exponential moving average of the target networks is updated with an initial $\tau$ of 0.996, and following the update defined in Grill et al. (2020).

As we have two datasets of sessions, one for each country, we trained two models. In both cases the models were trained by running one epoch over the whole dataset. The datasets contained 12,421,493 pairs of products for site A and 7,978,613 for site B. Those pairs of items were obtained from a set of 300,000 users sessions for each site. To avoid a combinatorial explosion, we limited the pairs in each session by truncating each session to a maximum of 10 products. This however resulted in some problems when trying to directly compare to Meta-Prod2Vec as for the latter we use directly the full 300,000 sessions datasets. It is something to address in future releases. The batch size was of 4096. We used Stochastic Gradient Descent with a learning rate of 0.45, a momentum of 0.9 and a weight decay of 1e-5. The learning rate was scaled to the batch size following the parameters given in Grill et al. (2020), as well as a Cosine Anealing rating schedule as established by the BYOL work.

With the encoder model we trained a Logistic Regression classifier for maximum of 1000 iterations and the liblinear solver Fan et al. (2008) of the implementation in the Scikit-Learn Pedregosa et al. (2011) package (we leave the rest of the parameters in it's default values).

## A.2 BASELINES

The detailed baselines are the following:

**fastText** For classification tasks we use fastText's text categorization Joulin et al. (2016) engine. This is the main baseline working for that kind of downstream tasks in production environments.

**Bag-of-Words + LR** We did some preliminary experiments using bag-of-words with a linear classifier (i.e. logistic regression) but we decided to discard it as it consistently showed less performance than fastText.

**Text CNN** Both for the classification and regression tasks we are interested in how the encoder can perform if trained from zero (including the embeddings) for the specified task. We use a Text CNN with the same parameters as the encoder $f_\theta \circ t_\theta$. It is trained for a maximum of 25 epochs over the whole dataset with an early stopping of of 5 iterations without gaining in the validation data. For this case, each experiment in each training sample (as explained in the next session) is run 5 times under different random seeds and we report the mean of the results.

**BETO (Spanish BERT)** We evaluate our model using the Spanish version of BERT Devlin et al. (2018), named BETO Cañete et al. (2020), since our dataset is in Spanish. Like for our model, we only use the encoded version of the data, i.e. the embedding for the [CLS] token. We use the implementation of the Huggingface Library Wolf et al. (2019). We feed the encoded [CLS] to a logistic regression classifier with the same characteristics as the one used to evaluate our self-supervised model (described in the previous section). The reason for showing our results using BETO and not multilingual BERT is because some preliminary results showed that BETO consistently had at least the same performance of BERT and generally outperformed it, which makes sense because it is trained specifically on Spanish text.

**Meta-Prod2Vec + KNN** The Meta-Prod2Vec algorithm was used for comparison purposes, although unlike the rest of the baselines, it depends on metadata that might not be present (e.g. the id of an item or the category). It was used alongside a K Nearest Neighbor classifier or regressor, depending on the task. The value of K was 5. To train the embeddings for

this algorithm we used the full dataset sessions for each country, with the id of the item as the main vector and the title and category of the item as the metadata.

