# OpenReview forum: "Constraining Latent Space to Improve Deep Self-Supervised e-Commerce Products Embeddings for Downstream Tasks"
_ICLR.cc/2021/Conference — Reject_

### Official Review · AnonReviewer2 · 2020-10-21
**Minimal contribution or novelty**

**Rating:** 3
**Confidence:** 4

**Review:**

This paper proposed a way of product representation learning via BYOL framework. Background introduced, model explained, experiments conducted. I don't think this paper is good enough for acceptance. See detailed comments:

1. The writing. The model part is not clearly written, and neither is the experiments set-up.

2. The novelty. It seems it's a derivative work from BYOL for this particular product embedding application.

3. The sampling method. It is well known that sampling via browse data suffers from bias caused by the search engine. For example, two products are browsed together only if the search engine decides to show them to the shoppers together. The search engine quality will largely impact the quality of the sampled data.

4. Missing important literatures and comparisons. The browse product pairs are essentially the degree-2 connection of products on the query-product bipartite graph. There are vast amount of work for learning representations on such graph.

5. The experiment section is not convincing. By applying the embedding to specific problems doesn't justify the method's superiority.

6. The results about different losses. Results are illustrated yet no discussions or insights are provided. What's the motivation behind?

---

### Official Review · AnonReviewer3 · 2020-10-21
**BYOL for learning product representations**

**Rating:** 4
**Confidence:** 4

**Review:**

This paper studies representation learning in an e-commerce setting. In particular, the paper explores the use of the recently proposed Bootstrap Your Own Latent (BYOL) framework to learn product representations. Rather than using different views of the same entity (as is done in the image domain), different products within a single shopping session are used. Experiments show that the unsupervised BYOL alone does not perform well in this setting. To remedy this issue, the objective is modified by adding a supervised product category prediction loss. Experimental comparisons are made to several baselines on four downstream supervised learning tasks.

### Strengths
1. Much of the recent success in unsupervised representation learning has been in the CV and NLP domain. Studying the effectiveness of these techniques in other domains is an important practical research direction.
2. Experiments empirically confirm the benefit of the proposed modification to the BYOL objective.

### Weaknesses
1. The study of the BYOL framework in a new domain is a welcome contribution (contribution 3). Unfortunately, the experiments do not effectively isolate and study this topic due to the number of factors which change between baselines (architecture, objective, training data, etc). Notably missing are benchmarks for an equivalent architecture (TextCNN) trained with a more standard representation learning setup using a contrastive loss. Without such comparisons it is not possible to determine whether the improved representation learning abilities of the BYOL framework in the CV domain transfer to the e-commerce domain, or if the improved performance is the result of architectural and other experimental differences between baselines.
2. The paper does not justify the precise modification of the BYOL objective presented here. Would adding an explicit contrastive term, which would be advantageous in scenarios where supervised data is not available, have been less, equally, or more effective? Why predict $c_{p_j}$ from $p_i$ rather than $c_{p_i}$? What about predicting the categories of all products in the session in a multi-label manner, etc? The choices made in the paper should be explained and justified.
3. Experimental details are not clearly described. For example, I could not find a description of the difference between Site A and Site B. The number of unique products and users is not specified. Are the products that appear in the supervised tasks a subset of the products that appear in the session level data, or are there "cold-start" items which the model did not see during pre-training? Is the BERT model fine-tuned for the specific down stream supervised task, or are only the classifier parameters updates? The Appendix leads me to believe the the latter, "we only use the encoded version of the data", in which case this baseline seems less representative of best practices.
4. The Appendix mentions that the Meta-Prod2Vec baseline uses the full session dataset, that this caused "some problems" with direct comparison, and that it will be addressed in a "future releases." The nature of "some problems" should be clarified and possibly mentioned in the main paper. If results are not comparable they should be removed or fixed before publication.
5. The paper overstates contributions. The architecture used here (TextCNN) is quite standard, and does not represent a significant contribution alone (contributions 1 and 2).

### Recommendation
I vote for rejection. Although I believe understanding the effectiveness of BYOL in non-CV domains is an important research direction, I do not believe the current paper effectively addresses this topic due to issues with baselines and a lack of experimental details.

### Additional Info
The paper contains grammar mistakes which I found made reading difficult at times. Most are relatively minor and did not obscure from the overall message and content of the paper, and therefore did not contribute substantially to my assessment. Below are several examples from Section 5.1.

1. "As we can **observer**, in almost all plots" => "As we can **observe**, in almost all plots"
2. "the better performance is for the value of α of 0.9 **specially** at very low training samples" => "the better performance is for the value of α of 0.9 **especially** at very low training samples"
3. "the curve is **more steep** for cases" => "the curve is **steeper** for cases"

---

### Official Review · AnonReviewer1 · 2020-10-28
**Concerns about main hypothesis and novelty of the paper**

**Rating:** 3
**Confidence:** 4

**Review:**

The authors study the problem of representation learning of marketplace products to apply in downstream tasks. More specifically, the authors extend the work of BYOL with a new objective function by adding a cross-entropy objective. The main hypothesis of this paper is that different products of the same browsing session can be thought of as different augmentations of the same session.

Overall, the paper is clear and easy to follow. However, I have a few concerns and questions for authors.

Concerns:
The first concern is that the main hypothesis of this paper is not very clear to me. Why can different products of the same browsing session be thought of as different augmentations? I did not find clear explanations behind this hypothesis. This hypothesis is not validated in the experimental section, which leads that the hypothesis is not convincing to me.

The second concern is about novelty. The authors claim that the main contributions of the paper lie in their novel deep encoder architecture. However, the bi-encoder architecture is not novel, and the novelty of added cross-entropy is also limited.

Questions:
1. Why can different products be thought of as augmentations to each other? Have you considered cases 1) complimentary products or 2) the users want to purchase several non-related products and browse them at the same time. Have you analyzed the similarities among products in the same browsing session?

2. The sensitive nature of datasets can be understood. Have you considered conducting experiments on other public datasets for easy reproductions?

3. The gap between the performance of the proposed model and the state-of-the-art models is large. The authors claim that one of the advantages is that the inference of their model does not rely on GPU. Why do the authors think that reliance on GPU is a limitation? Moreover, considering the inference efficiency, how does the proposed model compare to other baselines?

---

### Official Review · AnonReviewer4 · 2020-10-28
**This paper proposes an interesting approach to generate latent space embeddings for e-commerce products to be used for other downstream tasks. The main contribution of the paper in this problem space is to eliminate the need to generate negative pairs which requires significant resources and bandwidth as a preprocessing stage or memory intensive training stage.**

**Rating:** 5
**Confidence:** 3

**Review:**

Overall, I vote for weak rejection. The idea of being able to learn good latent space embedding with less expensive resources is very beneficial and can have a lot of easy to use applications in e-commerce problem space. The input used in the paper of product title and category are usually widely available for such datasets. The concerns are detailed in the cons section and hopefully the authors can address my concerns in the rebuttal period.

######################################################################

Pros:
(1) The paper has good clarity and is mostly understandable. Interesting and useful enhancement to existing self-supervised learning work of BYOL for application in the new problem space of e-commerce.
(2) The optimization objectives are well defined and the hypothesis of adding the category-based loss to original BYOL loss is well supported by evaluation and experiments presented in the paper.
(3) The proposed method has good value and is supported by the authors’ extensive comparison experiments done with industry standard baselines like Meta-Prod2vec, fastText which are commonly used for such tasks of learning product embeddings.
(4) Detailed explanation of the proposed approach and the related work. Extensive experiments across two datasets.


######################################################################

Cons:
(1) The authors use AUCROC as a measure of performance for their comparisons with other baselines, since it is threshold invariant. However, some of their downstream tasks are classification problems (counterfeit and forbidden product detection) which are typically more susceptible to class imbalance and minimizing false positives might be more important. Can the authors provide some more metrics which gives more insight into the performance of the model in such cases or provide an explanation as to why it might be unnecessary?
(2) The BYOL model used by authors as a base architecture has a significant difference from authors, in that the BYOL tries to learn the representation on the same image (online and target projections are for the same image). But the authors use item pairs from items interacted within the session. But given the scope of the e-commerce listing type, users can interact with a lot of unrelated or complimentary items within the session. To overcome this, they had to add the category constraint, and their experiments prove that without it, the model becomes random. But what happens when the training sample has a lot of unrelated/complimentary items as +ve pairs? One suggestion for downstream classification could be to evaluate the model’s performance on the category prediction. Can the authors provide more clarification on product assortment in general and within the session? (eg: if the marketplace data has products from diverse categories in the same session)

---

### Decision · Program_Chairs · 2021-01-07
**Final Decision**

**Decision:**

Reject

**Comment:**

All reviews are somewhat below the acceptance threshold. The main concerns are in terms of lack of novelty, and that some of the paper's main claims are unsupported. Many of the criticisms are quite focused on specific details, but these seem significant enough to have been deal-breakers for this submission.